# Clinical characteristics and predictive models for hospitalized patients with COVID-19 combined with bacterial pneumonia

**Man Yuan, Mei Liang, Jian Xu, Da He, Yanfang Zhang, Xiaoran Li, Jinzhi He, Yang Yang, Zhiyong Zong\*, Junyan Qu** *

Center of Infectious Diseases, West China Hospital of Sichuan University, Chengdu, China

\* zongzhiy@scu.edu.cn (ZZ), qujunyan15647@wchscu.cn (JQ)

## Abstract

### Background

This study aimed to analyze the clinical characteristics of hospitalized patients with COVID-19 combined with bacterial pneumonia and establish a predictive model to assist clinicians in the differential diagnosis and evaluation of the effectiveness of antibiotic treatment.

### Methods

In this retrospective study, we collected clinical, biochemical, imaging, and micro-biological data from hospitalized patients with COVID-19 admitted to our hospital between December 1, 2022, and February 7, 2023. Univariate and multivariate analyses revealed independent risk factors for bacterial pneumonia in COVID-19 patients. Model performance was assessed via the area under the curve (AUC).

### Results

A total of 5358 COVID-19 patients were screened, and data from 1794 patients were ultimately included; 1386 patients had concomitant bacterial pneumonia (77.3%), whereas 408 patients served as controls (22.7%). Among COVID-19 patients, those with concomitant bacterial pneumonia had lower levels of albumin, hemoglobin, and lymphocyte ratio, along with higher levels of procalcitonin, globulin, glucose, urea, white blood cell count, and neutrophil ratio, than patients without bacterial pneumonia. Sputum cultures identified *Acinetobacter baumanii*, *Klebsiella pneumoniae*, and *Pseudomonas aeruginosa* as the top three bacterial species. A predictive model for the early detection of concomitant bacterial pneumonia in COVID-19 patients was developed via multivariate regression analysis, with an AUC of 0.850 (p < 0.001).

which permits unrestricted use, distribution, and reproduction in any medium, provided the original author and source are credited.

**Data availability statement:** Our research data includes sensitive or confidential information such as patient data. Requests for additional study-related data should be directed to the Ethics Committee of West China Hospital, Sichuan University, via email at hxlcyjglb@163.com.

**Funding:** The author(s) received no specific funding for this work.

**Competing interests:** The authors have declared that no competing interests exist.

## Conclusion

These findings provide valuable insights for clinicians in the early diagnosis of bacterial pneumonia in COVID-19 patients, which may facilitate timely intervention and treatment.

## 1. Introduction

Since its emergence in late 2019, the COVID-19 pandemic has placed an extraordinary burden on global healthcare systems. It is caused by severe acute respiratory syndrome coronavirus 2 (SARS-CoV-2) and manifests with a wide range of respiratory symptoms, often leading to pneumonia and acute respiratory distress syndrome in severe cases [1,2]. The transition of COVID-19 has moved from a pandemic to an endemic state in recent years, akin to seasonal influenza which projects asynchronous surges of SARS-CoV-2 in different regions [3]. Hospitalized COVID-19 patients are prone to bacterial infections, which significantly exacerbate disease severity and complicate patient management. Bacterial pneumonia is associated not only with increased mortality but also with prolonged hospital stays and increased antibiotic use [4,5]. The phenomenon of bacterial coinfections is not unique to COVID-19 and has been observed in other viral respiratory infections. Viral infections, such as those caused by influenza and respiratory syncytial virus, are well-documented to predispose individuals to bacterial infections, which can exacerbate disease severity and increase mortality rates [6,7]. However, because the clinical manifestations and imaging findings of COVID-19 patients with bacterial pneumonia are similar to those of COVID-19 pneumonia patients, distinguishing them is difficult. Therefore, early identification of COVID-19 patients with bacterial pneumonia and effective antibacterial infection treatment are very important to reduce the severity of COVID-19 and promote the rational use of antibiotics.

Several studies have explored bacterial infections in COVID-19 patients, identifying risk factors such as advanced age, comorbidities, and prolonged mechanical ventilation [8–10]. Various bacterial species have been frequently implicated in these infections, and their presence further complicates the clinical course. Although various diagnostic markers, such as elevated levels of procalcitonin (PCT) and abnormal C-reactive protein (CRP), are associated with bacterial infections in COVID-19 patients, their specificity remains limited [11,12]. In recent years, predictive models based on clinical and biochemical parameters have gained increasing attention and have emerged as important tools for the early identification of high-risk hospitalized patients [13]. Although research is still ongoing, the integration of multiple diagnostic factors to accurately identify bacterial pneumonia in COVID-19 patients remains underexplored. Currently, most methods focus on individual risk factors, but few models combine clinical and laboratory data to improve diagnostic accuracy. The development of a comprehensive predictive model to identify COVID-19 patients with concurrent bacterial infections early is crucial for guiding antibiotic therapy, reducing unnecessary treatment, and improving patient outcomes.

This study aims to address this research gap by developing a predictive model for bacterial pneumonia in hospitalized COVID-19 patients. Through a retrospective analysis of clinical and biochemical data from a large cohort of patients, we aimed to identify independent risk factors and build a predictive model for early diagnosis. The model's performance was assessed via the area under the curve (AUC), providing a reliable tool for clinicians to differentiate bacterial infections from viral pneumonias and optimize antibiotic management in COVID-19 care.

## 2. Materials and methods

### 2.1. Study subjects

This study included patients with COVID-19 admitted to West China Hospital, Sichuan University (WCH, SCU) from December 1st, 2022, to February 27th, 2023. The hospital has 4300 beds and 54 clinical departments. In 2023, the WCH saw a staggering 8.78 million patients for outpatient and emergency visits and discharged 351,000 inpatients. Clinical data from all selected patients, including demographic characteristics, risk factors, clinical manifestations, laboratory test results, chest computed tomography (CT) images, treatments, and prognoses, were reviewed. The study protocol was approved by the Ethics Committee of WCH, SCU (Chengdu, Sichuan, China), following the ethical guidelines of the 1975 Declaration of Helsinki (approval number: 2023−30). Because the study did not directly interfere with the enrolled patients, the ethics committee waived informed consent. The data in this study were accessed for research purposes from March 29, 2023, and this retrospective study used a database from which the patients' identification information had been removed.

### 2.2. Enrollment criteria

We recruited all patients hospitalized with positive nucleic acid tests for SARS-CoV-2 before hospitalization admission. According to the Chinese Center for Disease Control and Prevention (CDC), the BA.5 subvariants was the dominant circulating variant (reached to 43%) at the time (around December 2022 to March 2023) when conducting this study in Chengdu, China. In this study, COVID-19 combined with bacterial pneumonia included COVID-19 coinfected with bacterial pneumonia and secondary bacterial pneumonia. Bacterial coinfection was defined as infection within 2 days of admission. Secondary infections were identified as bacterial infections that developed after admission for more than 2 days. Regarding the detection of pathogens in sputum, in addition to conventional sputum culture, some patients underwent additional pathogen detection methods, including respiratory virus and pathogenic bacterial nucleic acid testing, *Legionella* antigen testing, *Mycoplasma pneumoniae* antibody testing, *Chlamydia pneumoniae* antibody testing, and next-generation sequencing (NGS). According to the guidelines, combined bacterial pneumonia in COVID-19 patients is defined by the following criteria. The inclusion criteria were as follows: (1) respiratory symptoms such as cough, sputum production, and thick sputum; (2) wet rales on lung auscultation during physical examination; (3) elevated white blood count (WBC) and/or neutrophilia in routine blood tests; and (4) chest CT findings suggestive of bacterial pneumonia in addition to viral pneumonia (assessed independently by two radiologists) [14]. The exclusion criteria were as follows: (1) noninfectious factors such as lung cancer, heart failure, or pulmonary edema, which may cause chest CT changes; (2) chest CT changes caused by other infectious factors, such as fungi or tuberculosis; (3) positive sputum culture results considered to reflect colonization or contamination on the basis of clinical judgment. Two researchers, each holding the title of Associate Chief Physician or higher, independently assessed whether cases constituted contamination, colonization, or infection, adhering to the National Health Commission of the People's Republic of China (2001) Diagnostic Criteria for Nosocomial Infection (Trial) [15]. In instances of disagreement, a face-to-face discussion was conducted to reach a consensus. If a case was identified as contamination or colonization, the treatment plan remained unchanged, although relevant indicators and clinical symptoms were closely monitored; and (4) inability to differentiate between bacterial pneumonia and other conditions on the basis of clinical presentation, laboratory tests, or imaging findings. Controls without bacterial pneumonia

 

are characterized by the following characteristics: fever, predominantly dry cough, normal or reduced WBC count, normal PCT, absence of sputum or negative sputum cultures, no radiological evidence of bacterial pneumonia, and recovery from COVID-19 without antimicrobial therapy [16].

### 2.3. Data retrieval

We extracted clinical information from the hospital's electronic medical records database. Clinical data were collected within 3 days of coinfection or secondary infection with COVID-19. The following data were collected: comorbidities (including hypertension, diabetes, cardiovascular disease, chronic lung/renal diseases, immune system disease and malignancy), treatment (glucocorticosteroid, immunosuppressant, invasive mechanical ventilation, hematodialysis), demographic information (sex and age), PCT, biochemical markers (total bilirubin, alanine aminotransferase, aspartate aminotransferase, albumin, globulin, uric acid, glucose, urea, creatinine, the glomerular filtration rate, triglycerides, cholesterol, high-density lipoproteins, and low-density lipoproteins), electrolytes (sodium, potassium, calcium, and magnesium), blood cell tests (WBC count, hemoglobin, platelet count, lymphocyte ratio, and neutrophil ratio), antimicrobial exposure, and sputum and blood cultures, along with prognostic records.

### 2.4. Statistical analyses

All the statistical analyses were performed via SPSS version 22.0 (SPSS, Inc., Chicago, IL, USA). Continuous variables that did not follow a normal distribution are expressed as the median and interquartile range (IQR), whereas categorical variables are presented as frequencies and percentages. The $\chi^2$ test or Fisher's exact test was used to examine categorical variables such as sex and age. The Mann-Whitney U test was applied to analyze continuous variables. The forward Wald method was used to predict the model, which was based on the −2 log-likelihood ratio, Cox & Snell $R^2$, and Holmes goodness-of-fit, to filter the optimal logistic equation. Mortality predictors were determined by odds ratios (ORs) and 95% confidence intervals (CIs), which were calculated via multivariable binomial logistic regression analysis. A two-sided $P$ value of < 0.05 was considered statistically significant. Additionally, the corresponding AUC value was calculated to assess the model's discrimination ability.

## 3. Results

### 3.1. Clinical characteristics of patients

A total of 5,358 COVID-19 patients were screened in this study. Among them, 1,386 had combined bacterial pneumonia, whereas 408 did not. Notably, all 1,386 instances of combined bacterial pneumonia were categorized as secondary bacterial pneumonia. Table 1 shows the differences in the baseline characteristics, comorbidities, treatment, laboratory findings, and outcome parameters between the two groups. Compared with the control group, patients with combined pulmonary bacterial infections had lower levels of albumin, hemoglobin, and lymphocytes and higher PCT, globulin, glucose, and urea levels; WBC counts; and neutrophil ratios. The in-hospital mortality rate was 21.8% for patients with combined bacterial pneumonia, which was significantly higher than that of the control group (1.5%).

### 3.2. Common bacterial strains in COVID-19 patients

Out of 1060 positive culture samples (Fig 1a), the predominant isolates were *Acinetobacter baumanii* (244 cases, 23.02%), *Klebsiella pneumoniae* (204 cases, 19.25%), and *Pseudomonas aeruginosa* (90 cases, 8.49%). Among these, 833 sputum samples were positive for bacteria (Fig 1b), with *Acinetobacter baumanii* (210 cases, 25.21%), *Klebsiella pneumoniae* (165 cases, 19.81%), and *Pseudomonas aeruginosa* (83 cases, 9.96%) as the main isolates. Additionally, 168 blood cultures were positive, primarily for *Acinetobacter baumanii* (32 cases, 19.05%), *Klebsiella pneumoniae* (27 cases, 16.07%), and *Enterococcus faecium* (20 cases, 11.9%) (Fig 1c). There were only 13 cases in which the results of blood culture and respiratory tract culture were consistent.

**Table 1. Characteristics of COVID-19 patients with bacterial pneumonia.**

| Characteristics | Control (*n* = 408) | Bacterial pneumonia (*n* = 1386) | *P*-values |
|---|---|---|---|
| **Personal characteristics** | | | |
| Sex (male), *n* (%) | 228 (55.9) | 980 (70.7) | *0.000* |
| Age (years), median (IQR) | 59 (46, 73) | 73 (59, 83) | *0.000* |
| **Comorbidities** | | | |
| Hypertension, *n* (%) | 122 (29.9) | 619 (44.7) | *0.000* |
| Diabetes, *n* (%) | 71 (17.4) | 385 (27.8) | *0.000* |
| Cardiovascular disease, *n* (%) | 76 (18.6) | 460 (33.2) | *0.000* |
| Chronic lung diseases, *n* (%) | 62 (15.2) | 317 (22.9) | *0.001* |
| Chronic renal disease, *n* (%) | 72 (17.6) | 406 (29.3) | *0.000* |
| Immune system disease, *n* (%) | 33 (8.1) | 117 (8.4) | 0.821 |
| Malignancy, *n* (%) | 99 (24.3) | 188 (13.6) | *0.000* |
| **Treatment** | | | |
| Glucocorticosteroid, *n* (%) | 204 (50.0) | 1006 (72.6) | *0.000* |
| Immunosuppressant, *n* (%) | 43 (10.5) | 104 (7.5) | *0.049* |
| Invasive mechanical ventilation, *n* (%) | 57 (14.0) | 421 (30.4) | *0.000* |
| Hematodialysis, *n* (%) | 25 (6.1) | 65 (4.7) | 0.242 |
| **Laboratory findings** | | | |
| PCT (ng/mL), median (IQR) | 0.06 (0.05, 0.14) | 0.28 (0.09, 1.17) | *0.000* |
| TBiL (μmol/L), median (IQR) | 9.80 (6.82, 13.30) | 9.80 (6.80, 14.30) | *0.037* |
| ALT (IU/L), median (IQR) | 20 (13.04, 36.06) | 25 (15.79, 43.76) | *0.000* |
| AST (IU/L), median (IQR) | 24 (18, 36) | 32 (21, 52) | *0.000* |
| Albumin (g/L), median (IQR) | 37.75 (33.70, 41.20) | 32.65 (29.67, 35.80) | *0.000* |
| Globulin (g/L), median (IQR) | 24.70 (21.72, 27.80) | 25.70 (22.30, 29.60) | *0.000* |
| Uric Acid (μmol/L), median (IQR) | 281.5 (212, 366.75) | 261.50 (178.75, 382.25) | 0.050 |
| Glucose (mmol/L), median (IQR) | 5.55 (4.81, 7.05) | 7.32 (5.52, 10.38) | *0.000* |
| Sodium (mmol/L), median (IQR) | 139.10 (136.90, 141.20) | 138.80 (135.50, 142.10) | 0.977 |
| Potassium (mmol/L), median (IQR) | 3.98 (3.71, 4.28) | 4.03 (3.67, 4.46) | 0.051 |
| Calcium (mmol/L), median (IQR) | 2.18 (2.09, 2.27) | 2.06 (1.96, 2.17) | *0.000* |
| Magnesium (mmol/L), median (IQR) | 0.88 (0.81, 0.94) | 0.90 (0.81, 0.98) | *0.011* |
| Urea (mmol/L), median (IQR) | 5.30 (4, 7.57) | 8.90 (5.70, 16) | *0.000* |
| Creatinine (μmol/L), median (IQR) | 73 (59, 97.75) | 91 (68, 162.25) | *0.000* |
| eGFR (ml/min/1.73m$^2$), median (IQR) | 87.04 (65.63, 104.18) | 65.42 (33.71, 88.49) | *0.000* |
| Triglyceride (mmol/L), median (IQR) | 1.27 (0.94, 2.01) | 1.40 (1.03, 1.94) | 0.117 |
| Cholesterol (mmol/L), median (IQR) | 3.91 (3.23, 4.72) | 3.36 (2.67, 4.12) | *0.000* |
| HDL (mmol/L), median (IQR) | 1.06 (0.84, 1.25) | 0.88 (0.68, 1.14) | *0.000* |
| LDL (mmol/L), median (IQR) | 2.19 (1.66, 2.85) | 1.66 (1.11, 2.28) | *0.000* |
| WBC (×10$^9$/L), median (IQR) | 5.74 (4.27, 7.35) | 7.91 (5.13, 11.42) | *0.000* |
| Hb (g/L), median (IQR) | 120 (105, 136.75) | 111.50 (93, 128) | *0.000* |
| PLT (×10$^9$/L), median (IQR) | 194.50 (141.25, 257.50) | 159 (108.75, 222) | *0.000* |
| Lymphocyte ratio (%), median (IQR) | 19.15 (13, 28.40) | 8.20 (4.40, 14.62) | *0.000* |
| Neutrophil ratio (%), median (IQR) | 68.60 (58.92, 76.97) | 84.40 (75.10, 90.72) | *0.000* |
| **Prognosis** | | | |
| Recovery, *n* (%) | 402 (98.5) | 1084 (78.2) | *0.000* |
| Death, *n* (%) | 6 (1.5) | 302 (21.8) | |

IQR, interquartile range; PCT, procalcitonin; TBiL, total bilirubin; ALT, alanine aminotransferase; AST, aspartate aminotransferase; eGFR, estimated glomerular filtration rate; HDL, high-density lipoprotein; LDL, low density lipoprotein; WBC, white blood cell count; Hb, hemoglobin; PLT, platelet count.

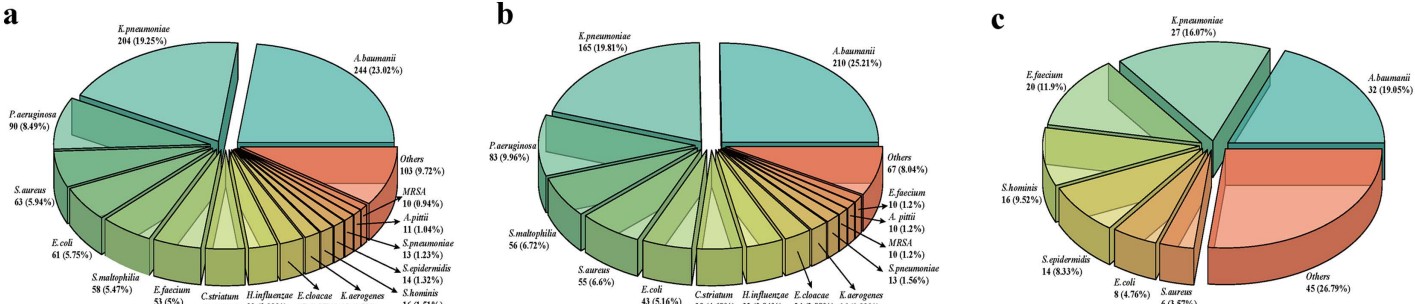

**Fig 1. Microbial etiology in COVID-19 patients with bacterial pneumonia. a.** Bacterial isolates from all clinical specimens. **b.** Bacterial isolates from sputum. **c.** Bacterial isolates from blood. *A.baumanii, Acinetobacter baumanii; K.pneumoniae, Klebsiella pneumoniae; P.aeruginosa, Pseudomonas aeruginosa; S.aureus, Staphylococcus aureus; E.coli, Escherichia coli; S.maltophilia, Stenotrophomonas maltophilia; E.faecium, Enterococcus faecium; C.striatum, Corynebacterium striatum; H.influenzae, Haemophilus influenzae; E.cloacae, Enterobacter cloacae; K.aerogenes, Klebsiella aerogenes; S.hominis, Staphylococcus hominis; S.epidermidis, Staphylococcus epidermidis; S.pneumoniae, Streptococcus pneumoniae; A.pittii, Acinetobacter pittii.*

### 3.3. Types of antibiotics utilized in treatment

The antimicrobial use strategies used to treat COVID-19 combined with bacterial pneumonia were also analyzed. Beta-lactamase inhibitors were the most commonly used drugs and were administered to 961 patients, followed by carbapenems, with 643 prescriptions, and quinolones, which were used in 430 patients. Additionally, cephalosporins were used in 349 patients, tetracyclines in 149 patients, polymyxins in 111 patients, and oxazolidinones in 88 patients (Fig 2). Among the patients, there were differences in the outcomes among patients who were prescribed cefoperazone-sulbactam, carbapenems, quinolones, cefmetazole sodium, cefoxitin sodium, tetracyclines, vancomycin, polymyxins and oxazolidinones (P < 0.05, Table 2).

### 3.4. Predictive models for COVID-19 combined with bacterial pneumonia

No significant differences were found between the two groups, and collinearity indices were excluded. To streamline the indicators, we have not incorporated specific comorbidities and treatment measures. Finally, sex, age, PCT, alanine aminotransferase, aspartate aminotransferase, high-density lipoprotein, glucose, blood calcium, blood magnesium, WBC count, platelet count, neutrophil percentage, albumin, and low-density lipoprotein were included in the regression analysis. We categorized the WBC count and PCT level into three groups. If the WBC count is $\geq 4 \times 10^9$/L and $< 8 \times 10^9$/L, it is classified as level 0; if the WBC count is $\geq 8 \times 10^9$/L and $< 17 \times 10^9$/L, it is classified as level 1; if the WBC count is $\geq 17 \times 10^9$/L or $< 4 \times 10^9$/L, it is classified as level 2. If the PCT level is $< 0.2$ ng/mL, it is classified as level 0; if the PCT level is $\geq 0.2$ ng/mL and $< 2$ ng/mL, it is classified as level 1; if the PCT level is $\geq 2$ ng/mL, it is classified as level 2. A multivariable regression analysis was subsequently performed to develop a predictive model for COVID-19 patients with concomitant pulmonary bacterial infection. The predictive model was as follows: predictive model = 1.234 * PCT classification + 0.339 * WBC classification + 0.053 * glucose + 0.034 * neutrophil ratio − 0.088 * albumin + 0.028 * age − 1.169 (Table 3). The AUC was calculated to determine the cutoff value for the predictive model. The results indicated that when the predictive model score exceeded 1.15, pulmonary bacterial coinfection was predicted, with a sensitivity of 77.6% and a specificity of 78.2% (Table 4). Finally, we compared the predictive abilities of the predictive model with those of PCT and WBC when used individually. Compared with PCT values, PCT classification, WBC values or WBC classification alone, our newly constructed predictive model demonstrated superior predictive ability (Fig 3; Table 4).

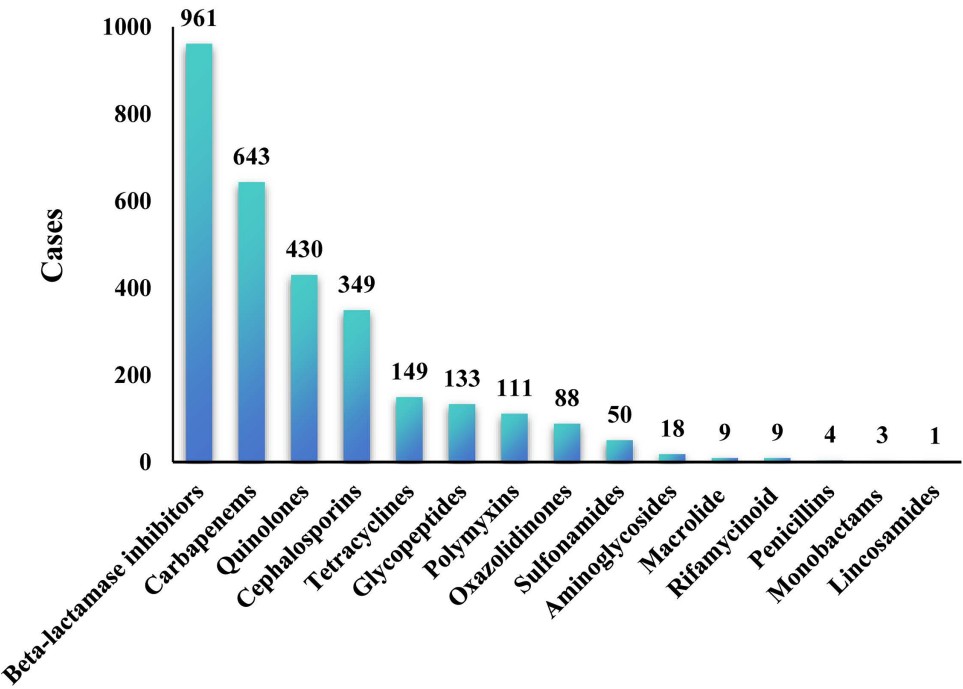

**Fig 2. Antibiotic use in COVID-19 patients with bacterial pneumonia.**

## 4. Discussion

Our research indicates that bacterial pneumonia significantly affects laboratory results and in-hospital mortality rates. The mortality rate among patients with bacterial pneumonia complications was 21.8%, whereas it was 1.5% in the control group. The elevated mortality rate observed in our study is similar to that reported in previous studies, which reported higher mortality rates in COVID-19 patients with bacterial infections than in those without [17]. These findings highlight the significant burden posed by concurrent bacterial infections in COVID-19 patients, aligning with previous reports identifying bacterial infections as crucial factors in the worsening of viral pneumonia, including SARS-CoV-2 cases [18,19].

Our study identified several key differences in baseline characteristics and laboratory findings between COVID-19 patients with and without bacterial infections. Patients with bacterial pneumonia presented with reduced levels of albumin, hemoglobin, and lymphocytes and elevated PCT, globulin, glucose, and urea levels; WBC counts; and neutrophil ratios. The alterations in these indicators reflect heightened inflammatory responses and immune dysregulation, which are commonly observed in bacterial infections. PCT, in particular, is widely recognized as a biomarker of bacterial infection, with elevated PCT levels correlating more with bacterial than with viral etiology in respiratory illness patients [20]. The elevated WBC count and neutrophil percentage further highlight bacterial infections, as these parameters typically increase during bacterial infections due to neutrophil recruitment and activation as part of the host immune response [21]. A study from Austin Bolker et al. revealed that PCT and WBC levels are significantly higher in respiratory bacterial coinfection patients than in patients without respiratory bacterial coinfection, which is similar to our findings [22]. Notably, significantly reduced albumin levels are detected in patients with concurrent pulmonary bacterial infections, as hypoalbuminemia is associated with bacterial infections and worse outcomes in patients with COVID-19 [23]. Albumin is a negatively regulated acute-phase reactant that decreases in response to infection, causing endothelial dysfunction and exacerbating fluid shifts, ultimately leading to adverse clinical outcomes [24]. The reduced hemoglobin levels and lymphopenia observed in the

**Table 2. Comparison of outcomes between patients with different antibiotics.**

| Antibiotic categories | Total (*n* = 1386) | Recovery (*n* = 1084) | Death (*n* = 302) | *P*-values |
|---|---|---|---|---|
| Beta-lactamase inhibitors, *n* (%) | 961 (100) | 739 (76.9) | 222 (23.1) | 0.075 |
| Piperacillin-tazobactam, *n* (%) | 580 (100) | 450 (77.6) | 130 (22.4) | 0.633 |
| Piperacillin-sulbactam, *n* (%) | 69 (100) | 60 (87) | 9 (13) | 0.071 |
| Cefoperazone-sulbactam, *n* (%) | 308 (100) | 227 (73.7) | 81 (26.3) | *0.030* |
| Ceftazidime-avibactam, *n* (%) | 4 (100) | 2 (50) | 2 (50) | 0.209 |
| Carbapenems, *n* (%) | 643 (100) | 413 (64.2) | 230 (35.8) | *0.000* |
| Ertapenem, *n* (%) | 17 (100) | 9 (52.9) | 8 (47.1) | *0.018* |
| Meropenem, *n* (%) | 397 (100) | 253 (63.7) | 144 (36.3) | *0.000* |
| Imipenem-Cilastatin, *n* (%) | 229 (100) | 151 (65.9) | 78 (34.1) | *0.000* |
| Quinolones, *n* (%) | 430 (100) | 370 (86.0) | 60 (14) | *0.000* |
| Moxiloxacin, *n* (%) | 318 (100) | 274 (86.2) | 44 (13.8) | *0.000* |
| Levofloxacin, *n* (%) | 112 (100) | 96 (85.7) | 16 (14.3) | *0.045* |
| Cephalosporins, *n* (%) | 349 (100) | 312 (89.4) | 37 (10.6) | *0.000* |
| Cefazolin Sodium, *n* (%) | 8 (100) | 7 (87.5) | 1 (12.5) | 0.695 |
| Cefaclor, *n* (%) | 4 (100) | 4 (100) | 0 | 0.582 |
| Cefprozil, *n* (%) | 1 (100) | 1 (100) | 0 | 1.000 |
| Cefotaxime Sodium, *n* (%) | 21 (100) | 19 (90.5) | 2 (9.5) | 0.196 |
| Cefmetazole Sodium, *n* (%) | 108 (100) | 95 (88) | 13 (12) | *0.011* |
| Cefoxitin Sodium, *n* (%) | 88 (100) | 82 (93.2) | 6 (6.8) | *0.000* |
| Ceftriaxone Sodium, *n* (%) | 55 (100) | 48 (87.3) | 7 (12.7) | 0.097 |
| Cefdinir, *n* (%) | 6 (100) | 5 (83.3) | 1 (16.7) | 1.000 |
| Ceftazidime, *n* (%) | 58 (100) | 51 (87.9) | 7 (12.1) | 0.067 |
| Tetracyclines, *n* (%) | 149 (100) | 87 (58.4) | 62 (41.6) | *0.000* |
| Minocycline Hydrochloride, *n* (%) | 2 (100) | 0 | 2 (100) | *0.047* |
| Tigecycline, *n* (%) | 147 (100) | 87 (59.2) | 60 (40.8) | *0.000* |
| Glycopeptides, *n* (%) | 133 (100) | 89 (66.9) | 44 (33.1) | *0.001* |
| Norvancomycin, *n* (%) | 11 (100) | 6 (54.5) | 5 (45.5) | 0.069 |
| Vancomycin, *n* (%) | 122 (100) | 83 (68) | 39 (32) | *0.004* |
| Polymyxins, *n* (%) | 111 (100) | 65 (58.6) | 46 (41.4) | *0.000* |
| Polymyxini B Sulphas, *n* (%) | 111 (100) | 65 (58.6) | 46 (41.4) | *0.000* |
| Oxazolidinones, *n* (%) | 88 (100) | 56 (63.6) | 32 (36.4) | *0.001* |
| Linezolid, *n* (%) | 88 (100) | 56 (63.6) | 32 (36.4) | *0.001* |

bacterial pneumonia group may reflect the combined effects of COVID-19-associated inflammation and infections on immune and hematologic function, further increasing the likelihood of adverse clinical outcomes.

The most common bacterial pathogens identified in the sputum cultures in this study were *Acinetobacter baumanii (25.21%), Klebsiella pneumoniae (19.81%), and Pseudomonas aeruginosa (9.96%).* These findings are consistent with previous studies that emphasized the predominance of gram-negative bacteria, particularly *Pseudomonas aeruginosa* and *Klebsiella pneumoniae*, in secondary infections in COVID-19 patients [18]. In addition, Musuuza et al. reported that bacterial coinfections and superinfections occurred in approximately 8% and 20% of hospitalized COVID-19 patients, respectively, with common pathogens, including *Klebsiella pneumoniae*, *Acinetobacter spps*, *Staphylococcus aureus* and *Streptococcus pneumoniae* [19]. The high prevalence of these nosocomial pathogens raises concerns about hospital-acquired infections, which may complicate the course of critically ill patients, especially those requiring prolonged mechanical ventilation or invasive procedures. *Pseudomonas aeruginosa* and *Klebsiella pneumoniae* are known for colonizing

**Table 3. Effect values of the indicators of the prediction model of bacterial pneumonia in COVID-19 patients.**

| Factor | β | Odds ratio | 95% CI | | P-values |
| --- | --- | --- | --- | --- | --- |
| | | | Lower | Upper | |
| Age | 0.028 | 1.029 | 1.021 | 1.036 | *0.000* |
| PCT classification | 1.234 | 3.434 | 2.622 | 4.497 | *0.027* |
| WBC classification | 0.339 | 1.404 | 1.186 | 1.662 | *0.000* |
| Glucose | 0.053 | 1.054 | 1.009 | 1.102 | *0.019* |
| Neutrophil ratio | 0.034 | 1.034 | 1.025 | 1.044 | *0.000* |
| Albumin | −0.088 | 0.916 | 0.893 | 0.939 | *0.000* |
| Constant | −1.169 | 0.311 | – | – | 0.087 |

PCT, procalcitonin; WBC, white blood cell count; CI, confidence interval.

**Table 4. The predictive value of different forecasting indicators in COVID-19 complicated with bacterial pneumonia.**

| Factor | Cut off | Sensitivity (%) | Specificity (%) | ROC | 95% CI | P-values |
| --- | --- | --- | --- | --- | --- | --- |
| PCT | 0.10 | 71.4 | 71.3 | 0.770 | 0.750-0.790 | *<0.0001* |
| PCT classification | 0 | 58.1 | 82.1 | 0.715 | 0.693-0.735 | *<0.0001* |
| WBC | 7.49 | 53.7 | 77.5 | 0.666 | 0.644-0.688 | *<0.0001* |
| WBC classification | 0 | 63.8 | 58.6 | 0.587 | 0.564-0.610 | *<0.0001* |
| Predictive model | 1.15 | 77.6 | 78.2 | 0.850 | 0.833-0.866 | *<0.0001* |

PCT, procalcitonin; WBC, white blood cell count; CI, confidence interval; ROC, area under the curve.

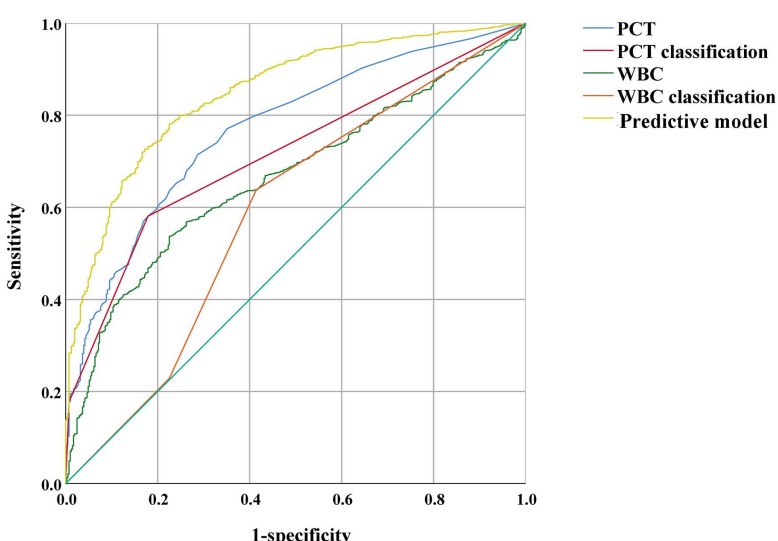

**Fig 3. ROC curve of different forecasting indicators for COVID-19 combined with bacterial pneumonia.** PCT, procalcitonin; WBC, white blood cell count; ROC, area under the curve.

the respiratory tract and causing severe pneumonia, especially in patients with underlying lung disease or immunocompromised patients [25–27]. Similarly, the detection of MRSA underscores the challenge of treating bacterial infections in COVID-19 patients, as MRSA infections require targeted antibiotic therapies, and delays in appropriate treatment may

lead to poor outcomes [28]. Interestingly, our study also revealed bacterial infections in sites other than the lungs, including bloodstream. Notably, the most prevalent pathogens in blood cultures were *Acinetobacter baumanii* (19.05%), *Klebsiella pneumoniae* (16.97%), and *Enterococcus faecium* (11.9%). These findings suggest that bacterial dissemination may occur in COVID-19 patients, particularly those with severe disease, as systemic inflammatory responses and immune dysregulation may predispose them to infections at multiple sites. The occurrence of bloodstream infections further emphasizes the complexity of managing bacterial infections in COVID-19 patients, highlighting the need for comprehensive diagnostic and therapeutic approaches.

This study reveals patterns of antibiotic use among COVID-19 patients and highlights the importance of beta-lactamase inhibitors and carbapenems in managing bacterial infections. In particular, during the COVID-19 pandemic, bacterial infections became more prevalent, largely driven by gram-negative and drug-resistant strains. The use of beta-lactamase inhibitor antibiotics (961 cases) ranked first, receiving considerable attention for their role in treating COVID-19, particularly in addressing lower respiratory tract infections. Moreover, the high usage rate of carbapenem antibiotics (643 cases) underscores the widespread presence of multidrug-resistant bacteria, such as carbapenem-resistant *Klebsiella pneumoniae* and *Acinetobacter baumanii*. These antibiotics are considered the last line of defense in the treatment of COVID-19 infections; however, their frequent use may facilitate the further spread of resistant strains. For example, a study demonstrated a significant correlation between carbapenem use and the emergence of drug-resistant bacteria in COVID-19 patients [29]. Additionally, the use of tetracyclines (149 cases), polymyxins (111 cases), and oxazolidinones (88 cases) highlights their importance in treating severe drug-resistant bacterial infections. Polymyxins are primarily used to treat carbapenem-resistant gram-negative bacterial infections, such as *Acinetobacter baumanii* and *Pseudomonas aeruginosa*, especially in critically ill COVID-19 patients [30]. In conclusion, the antibiotic usage patterns observed in this study underscore the challenges posed by bacterial infections during the COVID-19 pandemic, particularly the threat of multidrug-resistant bacteria. Further research is needed to evaluate the impact of these prescriptions on patient outcomes and resistance and to advocate for stricter antibiotic stewardship strategies to minimize unnecessary use.

To aid in the early identification of COVID-19 patients at risk for pulmonary bacterial coinfections, we developed a predictive model based on key clinical and laboratory parameters, including PCT, WBC count, glucose, albumin, and age. The selection of the cut-off values for PCT and WBC levels was made to make the model more convenient to use, especially in areas with scarce medical resources. Our model demonstrated good predictive performance, with an AUC of 0.85, and outperformed the use of individual biomarkers, such as PCT or WBC count alone. This finding is particularly relevant, as it offers clinicians a more comprehensive and precise tool for assessing the risk of bacterial pneumonia in COVID-19 patients. While PCT has been widely used as a biomarker for bacterial infections, its diagnostic accuracy can vary, particularly in patients with viral infections such as COVID-19, where inflammation may confound PCT levels [31]. By incorporating additional variables such as WBC count, glucose, and albumin, our model enables a more detailed assessment of the patient's risk profile, potentially allowing for earlier identification of at-risk patients and more timely initiation of appropriate antibiotic therapy. Several studies have highlighted the prevalence and impact of bacterial coinfections in patients with viral respiratory illnesses, including COVID-19 [18,32]. Our findings are consistent with the literature, which has shown that bacterial infections significantly increase morbidity and mortality in patients with viral pneumonia. Our research adds to the literature by providing a predictive model specifically designed to identify COVID-19 patients at risk for bacterial pneumonia. Past studies have focused on identifying risk factors for bacterial coinfection, primarily using a single PCT or CRP indicator [11,12]. Compared with the previous model, which includes fewer or more complicated indicators, our model can be used for early diagnosis of clinically easy-to-obtain indicators. It can be used in multiple time periods, especially when there is suspicion of secondary infection. Therefore, our model represents a novel contribution to the field by providing a practical tool for clinicians to enhance patient management and improve outcomes.

Despite the transition of the COVID-19 pandemic to an endemic phase, the virus is expected to persist for an extended period. The occurrence of COVID-19 complicated by bacterial pneumonia will continue to be a prevalent clinical scenario.

The findings of this study hold significant clinical relevance for the early diagnosis of COVID-19 complicated by pneumonia and for reducing mortality in these patients.

Our study also has several limitations. As a single-center retrospective study, there may be selection bias that limits the generalizability of the results. However, this large sample study provides useful information. In addition, we did not include bacterial antimicrobial resistance, which was analyzed in combination with the selection of antimicrobials and clinical outcomes. The focus of this study was to establish a prediction model for COVID-19 combined with bacterial pneumonia, and relevant studies on antimicrobial resistance should be conducted in the future. Finally, this study did not include research on patients with upper respiratory tract infections (URTI), sepsis, or similar conditions. Future research will focus on these populations to further validate the predictive model and assess its applicability to infections in other parts.

## 5. Conclusions

In conclusion, our study demonstrated that COVID-19 patients have a significant increase in mortality due to bacterial pneumonia. *Acinetobacter baumanii*, *Klebsiella pneumoniae*, and *Pseudomonas aeruginosa* are the top three bacterial species in sputum cultures. Beta-lactamase inhibitors are the most commonly used antibiotics. The novel predictive model may be helpful for the early detection of bacterial pneumonia in COVID-19 patients.

## Author contributions

**Data curation:** Man Yuan, Mei Liang, Jian Xu, Da He, Yanfang Zhang, Xiaoran Li, Jinzhi He, Yang Yang.

**Writing – original draft:** Man Yuan.

**Writing – review & editing:** Zhiyong Zong, Junyan Qu.

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
