## [Decision Letter · Decision Letter 0]

24 Jul 2025

Dear Dr. Qu,

We look forward to receiving your revised manuscript.

Kind regards,

Padmapriya P Banada, PhD

Academic Editor

PLOS ONE

Journal Requirements:

2. Please note that PLOS ONE has specific guidelines on code sharing for submissions in which author-generated code underpins the findings in the manuscript. In these cases, we expect all author-generated code to be made available without restrictions upon publication of the work. 

Please review our guidelines at https://journals.plos.org/plosone/s/materials-and-software-sharing#loc-sharing-code and ensure that your code is shared in a way that follows best practice and facilitates reproducibility and reuse.

4. In this instance it seems there may be acceptable restrictions in place that prevent the public sharing of your minimal data. However, in line with our goal of ensuring long-term data availability to all interested researchers, PLOS’ Data Policy states that authors cannot be the sole named individuals responsible for ensuring data access (http://journals.plos.org/plosone/s/data-availability#loc-acceptable-data-sharing-methods).

**Additional Editor Comments:**

Thank you for your patience as this manuscript took a while to get reviewed. Please see the reviewers comments. I hope that you find them useful to improve the manuscript and I would be happy to recommend this paper for publication provided you answer all of the reviewers concerns. please ensure that you incorporate the requested changes in the revised version.

Reviewers' comments:

Reviewer's Responses to Questions

**Comments to the Author**

1. Is the manuscript technically sound, and do the data support the conclusions?

Reviewer #1: Yes

Reviewer #2: Yes

2. Has the statistical analysis been performed appropriately and rigorously?

Reviewer #1: Yes

Reviewer #2: Yes

3. Have the authors made all data underlying the findings in their manuscript fully available?

Reviewer #1: Yes

Reviewer #2: Yes

4. Is the manuscript presented in an intelligible fashion and written in standard English?

Reviewer #1: Yes

Reviewer #2: Yes

Reviewer #1: The idea of the manuscript is good, although the COVID-19 as the main topic has switched into common disease.

Please see the upload files for detailed of comment specific related the manuscript, and author need to clarify each of the question clearly.

Reviewer #2: The study is devoted to an essential problem – finding a way to predict the likelihood of bacterial pneumonia in patients with COVID-19. It is known that there are no reliable diagnostic criteria, which leads to hypo- and hyperdiagnosis and, accordingly, unjustified prescription of antibiotics.

However, despite the relevance of the problem raised by the paper, there are significant questions on methodology and the results that I would ask to clarify.

Materials and Methods

1. Selection of the population to be analysed

Why did the authors choose to combine bacterial pneumonia as coinfections and secondary infection (namely nosocomial pneumonia) into one group? These are different cohorts of patients, with different risk factors on the development of pneumonia, different aetiology, prognosis, and decisions to be made. It is well known that if we select good studies, bacterial coinfections in COVID-19 remain rare, but secondary superinfections, including nosocomial pneumonia, are frequent and indeed significantly worsen the prognosis.

2. Inclusion/exclusion criteria

- One of the exclusion criteria requires clarification:

3) positive sputum culture results considered to reflect colonisation or contamination on the basis of clinical judgement. What does clinical judgement mean in this case to assess the results as insignificant? What specific criteria were used?

- Selection of the control group: why the absence of sputum (it is also quite common in bacterial pneumonia, especially at the onset of the disease) as well as recovery without AB therapy allowed to exclude bacterial co-/superinfection?

3. When choosing bacterial pneumonia, you have relied only on sputum culture, which allow the detection of only typical bacterial pathogens. What was the case with the detection

of routinely non-culturable but significant community-acquired bacteria such as M. pneumoniae, L. pneumophila?

4. You developed the pneumonia model, why did you evaluate the urine culture data?

Results

1. Why, out of 5,358 COVID-19 patients, you could only include 1,386 in the study group and 408 in the control group. Explain the reasons.

2. Please explain what was the proportion of bacterial co-infection and secondary/ superinfection in the study group? This is important in terms of understanding the structure of the pathogens that are presented in the article.

3. When assessing the results of sputum culture, was the quality of samples evaluated, in particular, were the results of Gram staining taken into account?

4. How were cases of bacteriemia interpreted? Were the isolated microorganisms considered to be pathogens of pneumonia? Did the blood cultures and respiratory sample cultures match?

5. Why does the article provide data on AB prescription if the quality of their usage is not further analyzed? The prescriptions are given only for AB groups, including beta-lactams, that contain drugs with a fundamentally different spectrum of activity.

6. The choice of cut-off points for WBCs and PCT level needs to be explained.

Discussion

1. Please explain what treatment patients received for COVID-19. You claim that

SARS-CoV-2 was a major risk factor for bacterial co-/superinfections. However, treatment, particularly the use of glucocorticosteroids and immunosuppressants, as well as the presence of various invasive devices in severe cases, are equally important risk factors for bacterial superinfections.

2. It would be useful to add to the discussion at what stage you propose to use the developed model for predicting bacterial pneumonia – upon admission, after 48 hours, or at other times?

3. The section on research limitations must be expanded; in particular, it does not mention that bacterial pathogens were identified solely on the basis of sputum culture.

Conclusion

You conclude that bacterial infection in COVID-19 has an adverse effect on prognosis. Was the risk of mortality adjusted in the study and control groups for confounding factors such as age and comorbidities, which are themselves associated with a poorer prognosis?

**Do you want your identity to be public for this peer review?** For information about this choice, including consent withdrawal, please see our Privacy Policy

Reviewer #1: **Yes: ** Adhi Kristianto Sugianli

Reviewer #2: No

---

## [Author Response · Author response to Decision Letter 1]

13 Oct 2025

Point by Point Response to Reviewers’ Comments

Journal Requirements:

Author response: Yes.

2. Please note that PLOS ONE has specific guidelines on code sharing for submissions in which author-generated code underpins the findings in the manuscript. In these cases, we expect all author-generated code to be made available without restrictions upon publication of the work.

Please review our guidelines at https://journals.plos.org/plosone/s/materials-and-software-sharing#loc-sharing-code and ensure that your code is shared in a way that follows best practice and facilitates reproducibility and reuse.

Author response: Our article does not contain any code.

Author response: The data contain potentially identifying or sensitive patient information, and the Research Ethics Committee has imposed restrictions on them. Requests for additional study-related data should be directed to the Ethics Committee of West China Hospital, Sichuan University, via email at hxlcyjglb@163.com.

4. In this instance it seems there may be acceptable restrictions in place that prevent the public sharing of your minimal data. However, in line with our goal of ensuring long-term data availability to all interested researchers, PLOS’ Data Policy states that authors cannot be the sole named individuals responsible for ensuring data access (http://journals.plos.org/plosone/s/data-availability#loc-acceptable-data-sharing-methods).

Author response: The data contain potentially identifying or sensitive patient information, and the Research Ethics Committee has imposed restrictions on them. Requests for additional study-related data should be directed to the Ethics Committee of West China Hospital, Sichuan University, via email at hxlcyjglb@163.com. The datasets are available from the Ethics Committee upon reasonable request.

Author response: Yes.

Author response: We have reviewed our reference list to ensure that it is complete and correct.

Author response: The reviewer's comments do not cover the above-mentioned matters.

Additional Editor Comments:

Reviewer #1: The idea of the manuscript is good, although the COVID-19 as the main topic has switched into common disease.

Please see the upload files for detailed of comment specific related the manuscript, and author need to clarify each of the question clearly.

1. The title is clear and well-described the purpose of the study outcome among the population.

Author response 1#: Thank you very much for your recognition.

2. In the Introduction: the rationale observing the clinical characteristics is clear enough, however since the COVID-19 has moved to endemic state, similar with seasonal influenza, it needs more elaborate argumentation with present situation. Please elaborate with other situation which mimic with COVID-19 with bacteria infection, that common occur in healthcare facilities.

Author response 1#: Thank you very much for your suggestion. According to your advice, we added that “The transition of COVID-19 has moved from a pandemic to an endemic state in recent years, akin to seasonal influenza which projects asynchronous surges of SARS-CoV-2 in different regions”in the introduction section. Besides, we added the elaborate with other situation which mimic with COVID-19 with bacteria infection, that common occur in healthcare facilities. The phenomenon of bacterial coinfections is not unique to COVID-19 and has been observed in other viral respiratory infections. Viral infections, such as those caused by influenza and respiratory syncytial virus, are well-documented to predispose individuals to bacterial infections, which can exacerbate disease severity and increase mortality rates. The revision located on Page 4, line 52-55 and line 58-62, marked red.

3. May the predictive model of the study applied with other viral bacterial infection, e.g. bacterial + viral sepsis? If yes, please add information into the manuscript, especially into the introduction.

Author response 1#: Thank you very much for your suggestion. We recognize that the predictive model of the study applied with other viral bacterial infection could provide more robust support for our conclusions. Consequently, we plan to incorporate a validation cohort regarding bacterial + viral sepsis in our future research.

4. In the method section:

- The immune response may be altered due to comorbidities in the population of COVID-19, however, there is no information about the screening of comorbidities among the population, as the enrollment criteria.

Author response 1#: Thank you very much for your suggestion. According to your advice, we added the comorbidities in the Data Retrieval section. The following data were collected: comorbidities (including hypertension, diabetes, cardiovascular disease, chronic lung/renal diseases, immune system disease, and malignancy). The revision located on Page 8, line 146-147, marked red. Table 1 shows the comorbidities. The revision located on Page 10.

- The population was taken between Dec 2022 — Feb 2023, How does the impact of the virus variant to co-infection/secondary bacterial infection? As we know that the shifting of variant play role during the pandemic, which implies to the clinical characteristics and disease outcome.

Author response 1#: Thank you very much for your suggestion. Our hospital tested the SARS-CoV-2 by RT-PCR test, but no subtype analysis was conducted. However, according to the Chinese Center for Disease Control and Prevention (CDC), the BA.5 subvariants was the dominant circulating variant (reached to 43%) at the time (around December 2022 to March 2023) when conducting this study in Chengdu, China. We added above information in the Enrollment criteria. The revision located on Page 6, line 108-111, marked red.

5. Result section 3.2: The paragraph written is difficult to follow, author need simplify and highlight the important finding of the study.

Author response 1#: Thank you very much for your suggestion. We modified it, and simplify and highlight the important finding of the study. Out of 1060 positive culture samples (Fig. 1a), the predominant isolates were Acinetobacter baumanii (244 cases, 23.02%), Klebsiella pneumoniae (204 cases, 19.25%), and Pseudomonas aeruginosa (90 cases, 8.49%). Among these, 833 sputum samples were positive for bacteria (Fig. 1b), with Acinetobacter baumanii (210 cases, 25.21%), Klebsiella pneumoniae (165 cases, 19.81%), and Pseudomonas aeruginosa (83 cases, 9.96%) as the main isolates. Additionally, 168 blood cultures were positive, primarily for Acinetobacter baumanii (32 cases, 19.05%), Klebsiella pneumoniae (27 cases, 16.07%), and Enterococcus faecium (20 cases, 11.9%) (Fig. 1c). There were only 13 cases in which the results of blood culture and respiratory tract culture were consistent. The revision located on Page 12-13, line 192-211, marked red.

6. The Figure 2 describes the stratification of patient recovery and death, based on antibiotic. What is the purposes to stratified using this approach?

Author response 1#: Thank you very much for your suggestion. The stratification of patient recovery and death based on antibiotic use is a critical approach in managing pneumonia. The primary purpose of stratifying patients in this manner is to optimize antibiotic therapy, ensuring that patients receive the most appropriate treatment based on their specific clinical conditions and the characteristics of the pathogens involved.

7. PREDICTIVE MODEL abbreviation seems confusing to read, may be need other abbreviation or synonym, e.g. predictive model, etc.

Author response 1#: Thank you very much for your suggestion. We modified “PREDICTIVE MODEL abbreviation” to “predictive model”. The revision located on Page 17, line 259-260, line 263, line 265, line 267 and Table 4, marked red.

8. In the discussion, there is no information about the further implication of this study to the recent situation since the COVID-19 has been turned into endemic phase. As mentioned earlier, does all the predictive model applies to other URTI or severe immune response disease, e.e.g sepsis?

Author response 1#: Thank you very much for your suggestion. Despite the transition of the COVID-19 pandemic to an endemic phase, the virus is expected to persist for an extended period. The occurrence of COVID-19 complicated by bacterial pneumonia will continue to be a prevalent clinical scenario. The findings of this study hold significant clinical relevance for the early diagnosis of COVID-19 complicated by pneumonia and for reducing mortality in these patients. The revision located on Page 24, line 391-395, marked red.

However, this study did not include research on patients with upper respiratory tract infections (URTI), sepsis, or similar conditions. Future research will focus on these populations to further validate the predictive model and assess its applicability to infections in other parts. We have added the above limitations in the Discussion section. The revision located on Page 24-25, line 402-405, marked red.

Reviewer #2: The study is devoted to an essential problem – finding a way to predict the likelihood of bacterial pneumonia in patients with COVID-19. It is known that there are no reliable diagnostic criteria, which leads to hypo- and hyperdiagnosis and, accordingly, unjustified prescription of antibiotics.

However, despite the relevance of the problem raised by the paper, there are significant questions on methodology and the results that I would ask to clarify.

Materials and Methods

1. Selection of the population to be analysed

Why did the authors choose to combine bacterial pneumonia as coinfections and secondary infection (namely nosocomial pneumonia) into one group? These are different cohorts of patients, with different risk factors on the development of pneumonia, different aetiology, prognosis, and decisions to be made. It is well known that if we select good studies, bacterial coinfections in COVID-19 remain rare, but secondary superinfections, including nosocomial pneumonia, are frequent and indeed significantly worsen the prognosis.

Author response 2#: Thank you very much for your suggestion. We carefully re-examined the data and found no co-infections. A total of 1,386 cases involved combined bacterial pneumonia, all of which were classified as secondary bacterial pneumonia. The revision located on Page 9, line 174-175, marked red.

2. Inclusion/exclusion criteria

- One of the exclusion criteria requires clarification:

3) positive sputum culture results considered to reflect colonisation or contamination on the basis of clinical judgement. What does clinical judgement mean in this case to assess the results as insignificant? What specific criteria were used?

- Selection of the control group: why the absence of sputum (it is also quite common in bacterial pneumonia, especially at the onset of the disease) as well as recovery without AB therapy allowed to exclude bacterial co-/superinfection?

Author response 2#: Thank you very much for your suggestion. Two researchers, each holding the title of Associate Chief Physician or higher, independently assessed whether cases constituted contamination, colonization, or infection, adhering to the National Health Commission of the People's Republic of China (2001) Diagnostic Criteria for Nosocomial Infection (Trial). In instances of disagreement, a face-to-face discussion was conducted to reach a consensus. If a case was identified as contamination or colonization, the treatment plan remained unchanged, although relevant indicators and clinical symptoms were closely monitored. The revision located on Page 7, line 129-135, marked red. References: The National Health Commission of the People's Republic of China (2001) Diagnostic Criteria

---

## [Editor Report · Decision Letter 1]

29 Oct 2025

Clinical characteristics and predictive models for hospitalized patients with COVID-19 combined with bacterial pneumonia

PONE-D-25-10475R1

Dear Dr. Qu,

We’re pleased to inform you that your manuscript has been judged scientifically suitable for publication and will be formally accepted for publication once it meets all outstanding technical requirements.

Kind regards,

Padmapriya P Banada, PhD

Academic Editor

PLOS ONE

Additional Editor Comments (optional):

Dear Authors,

Thank you for considering the suggestions and addressing the questions systematically. I am happy to recommend this manuscript for publication in PlosOne.
---

## [Editor Report · Acceptance letter]

PONE-D-25-10475R1

PLOS ONE

Dear Dr. Qu,

I'm pleased to inform you that your manuscript has been deemed suitable for publication in PLOS ONE. Congratulations! Your manuscript is now being handed over to our production team.

Kind regards,

on behalf of

Dr. Padmapriya P Banada

Academic Editor

PLOS ONE